# *Caenorhabditis elegans* as a Model to Study Manganese-Induced Neurotoxicity

**DOI:** 10.3390/biom12101396

**Published:** 2022-09-29

**Authors:** Airton C. Martins, Priscila Gubert, Jung Li, Tao Ke, Merle M. Nicolai, Alexandre Varão Moura, Julia Bornhorst, Aaron B. Bowman, Michael Aschner

**Affiliations:** 1Department of Molecular Pharmacology, Albert Einstein College of Medicine, Bronx, NY 10461, USA; 2Department of Biochemistry, Laboratório de Imunopatologia Keizo Asami, LIKA, Federal University of Pernambuco, Recife 50670901, Brazil; 3Postgraduate Program in Pure and Applied Chemistry, Federal University of Western of Bahia, Barreiras 47810059, Brazil; 4Des Moines University College of Osteopathic Medicine, Des Moines, IA 50312, USA; 5Stiles-Nicholson Brain Institute, Florida Atlantic University, Jupiter, FL 33458, USA; 6Food Chemistry, Faculty of Mathematics and Natural Sciences, University of Wuppertal, 42119 Wuppertal, Germany; 7TraceAge–DFG Research Unit on Interactions of Essential Trace Elements in Healthy and Diseased Elderly (FOR 2558), Berlin-Potsdam-Jena-Wuppertal, 14558 Nuthetal, Germany; 8MS4Life Laboratory of Mass Spectrometry, Health Sciences Postgraduate Program, São Francisco University, Bragança Paulista, São Paulo 12916900, Brazil; 9School of Health Sciences, Purdue University, West Lafayette, IN 47907, USA

**Keywords:** manganese, trace elements, alternative animal models, neurodegeneration, neurotoxicology, *C. elegans*

## Abstract

*Caenorhabditis elegans* (*C. elegans*) is a nematode present worldwide. The worm shows homology to mammalian systems and expresses approximately 40% of human disease-related genes. Since Dr. Sydney Brenner first proposed *C. elegans* as an advantageous experimental worm-model system for genetic approaches, increasing numbers of studies using *C. elegans* as a tool to investigate topics in several fields of biochemistry, neuroscience, pharmacology, and toxicology have been performed. In this regard, *C. elegans* has been used to characterize the molecular mechanisms and affected pathways caused by metals that lead to neurotoxicity, as well as the pathophysiological interrelationship between metal exposure and ongoing neurodegenerative disorders. Several toxic metals, such as lead, cadmium, and mercury, are recognized as important environmental contaminants, and their exposure is associated with toxic effects on the human body. Essential elements that are required to maintain cellular homeostasis and normal physiological functions may also be toxic when accumulated at higher concentrations. For instance, manganese (Mn) is a trace essential element that participates in numerous biological processes, such as enzymatic activities, energy metabolism, and maintenance of cell functions. However, Mn overexposure is associated with behavioral changes in *C. elegans*, which are consistent with the dopaminergic system being the primary target of Mn neurotoxicity. *Caenorhabditis elegans* has been shown to be an important tool that allows for studies on neuron morphology using fluorescent transgenic worms. Moreover, behavioral tests may be conducted using worms, and neurotransmitter determination and related gene expression are likely to change after Mn exposure. Likewise, mutant worms may be used to study molecular mechanisms in Mn toxicity, as well as the expression of proteins responsible for the biosynthesis, transport, storage, and uptake of dopamine. Furthermore, this review highlights some advantages and limitations of using the experimental model of *C. elegans* and provides guidance for potential future applications of this model in studies directed toward assessing for Mn neurotoxicity and related mechanisms.

## 1. *C. elegans* as an Alternative Model to Study Neurotoxicity

*Caenorhabditis elegans* (*C. elegans*) combines the simplicity of a small organism (reaching 1 mm when adult) with a notable biological conservancy with mammals, leveraging its use as a tool to solve crucial gaps in molecular biology [1,2,3,4], chemistry [5], biochemistry [6], toxicology [7,8,9], neuroscience [10], pathology [11], and genetics [12]. The easy genetic manipulation, including transgenics, mutants, and RNA interference (RNAi), which gave rise to multiple phenotypes highlights the biological dynamics in *C. elegans*.

The nematode is a soil-dwelling roundworm present worldwide. The wild-type (N2, Bristol) strain was established for research, from which many transgenic phenotypes were created. Worms can be grown in Petri dishes on nematode growth medium (NGM) nutritionally supplied with *Escherichia coli* (*E. coli*) OP50 at 20 °C. There is no need for CO_2_ supply or light/dark cycles. In fact the excess of light is injurious for *C. elegans* [13]. Laboratories interested in *C. elegans* research can start the colony by ordering various strains (wild-type and transgenics) and *E. coli* OP50 from Caenorhabditis Genetics Center (CGC). Compared to other models, *C. elegans* has a fast life cycle (3 days at 25 °C from egg to gravid adult), and its sexual forms comprise hermaphrodites and males, which represent 0.2% of the population. The self-fertilizing is controlled and can generate about 300 progenies per worm and 1000 progenies under crossed fertilizing. Males can be recognized by the shorter adult body size and tail shape differences, resembling an elegant fish fin [14].

The whole-body transparency of the nematode allows observation of intrabody structures and in vivo location of metal accumulation. Fluorescent protein tagging can be used to elucidate developmental process, neurotoxicity, fat storage, mitochondria integrity, and to characterize protein expression [15,16,17,18].

*Caenorhabditis* elegans sequencing has established 60–80% homology with the mammalian genome, with congruent basic biological functions similar to higher organisms, such as the digestive tract [19,20]. However, it has limitations for more complex studies, as some specific organs found in mammals are lacking, such as the eyes, lungs, and liver, as well as the circulatory and immunological systems [21,22].

Electron micrographs applied for the reconstruction of the shape of *C. elegans* have shown a constant number of cells (959 in adult hermaphrodites and 1031 in the adult male) and a stereotyped development [23]. The *C. elegans* nervous system includes 302 neurons in adult hermaphrodites and 385 in males due to the posterior mating circuit [24]. The evaluation of all connections from sensory input to end-organ output across the worm showed considerable connection differences between the sexes [25]. Functionally, the species has a similar nervous system to mammals, comprising well-defined neurotransmitters signaling, including dopamine (DA), serotonin (5-hydroxytryptamine, 5-HT), γ-aminobutyric acid (GABA), acetylcholine (ACh), and glutamate (Glu), which has turned it into a great model for neurotoxicity studies [26,27].

From a toxicological point of view, *C. elegans* is sensitive to many substances, including toxic metals, organic phosphates, and pesticides [28,29,30]. Accordingly, a screening was performed on ToxCast™ libraries of the Environmental Protection Agency (EPA), during which 968 chemicals were applied to *C. elegans* for new toxicological tests. The results obtained predicted developmental toxicity corroborating the results in rats, rabbits, and zebrafish [31].

Relevant parameters for toxicological screening can be evaluated in *C. elegans*, such as mortality, longevity, behavior, feeding, growth, and reproduction [32,33]. A relatively short incubation or treatment period is needed to examine the mortality or behavioral trials, ranging from a few minutes [26] to hours [5,7,8,34]. In this regard, neurodegeneration, reduction in growth, reproduction, feeding, locomotion, and worm lifespan have been evaluated under metal exposure, including with lead (Pb) [35], manganese (Mn) [36], titanium (Ti) [37], copper (Cu) [38], zinc (Zn) [39], gold (Au) [40], mercury (Hg) [41], iron (Fe) [42], and chrome (Cr) [43].

This review aims to highlight some essential advantages of using the experimental model *C. elegans* and to provide some application possibilities in studies of Mn neurotoxicity and related mechanisms. Moreover, we gathered the findings of the main studies related to Mn toxicity in *C. elegans*.

## 2. Manganese: Essential Element and Toxicity

Manganese is an essential metal for numerous enzymatic activities, energy metabolism, maintenance of cell functions, regulation of blood sugar, reproduction, and development [44]. The metal is the 12th most abundant element in the earth’s crust. It is distributed in the environment through natural processes, such as water and wind erosion, and anthropogenic activities, including ore mining and industrial manufacturing. Maintaining an adequate amount of Mn can be easily realized via dietary sources containing relatively high Mn levels, such as seeds and vegetables [45]. Thus, Mn deficiency is far less common than Mn overexposure. Although it is a nutritionally essential metal, Mn can be toxic if the levels are above approximately three times the physiological concentration (5.32–14.03 ng Mn/mg protein in the human brain) [46]. The concentration of Mn within the body can increase to toxic levels during the following scenarios: (1) during overexposure, which occur due to ingestion or inhalation of excess concentrations of Mn; (2) when there is an Fe deficiency, in which case transportation mechanisms shared by Fe and Mn facilitate the bioaccumulation of high concentrations of Mn; (3) when Mn excretion is impaired as a result of hepatic dysfunction [47,48].

The toxicity of Mn to humans was reported in workers in the mining and smelting industries and other occupational settings, such as welding, which generates high levels of Mn oxides. Manganese crosses the blood brain–barrier and eventually enters the human brain, where it disrupts neurotransmission and damages neurons important for motor functions, leading to a variety of clinical syndromes collectively known as manganism [49]. Manganism clinically resembles Parkinson’s disease (PD); however, the former involves the pathology in neurons of the globus pallidus as well as other brain areas, while PD is characterized by the progressive loss in dopaminergic neurons of the substantia nigra pars compacta (please refer to these reviews for further discussion of manganism [50,51,52,53]).

As an important industrial material, Mn has many uses, including in lithium-ion manganese oxide batteries, Mn alloys (e.g., ferromanganese alloys are used in welding rods), pigments, fertilizer, glass, and steel, a fundamental building component of the modern world. An organic Mn compound, methylcyclopentadienyl Mn tricarbonyl (MMT), was introduced as a gasoline additive that is currently used worldwide to increase the octane rating. Combustion of gasoline containing MMT (currently 8.3 mg Mn/L) releases manganese oxide into the atmosphere, causing a potential Mn-related health concerns for the human population, especially in heavily populated areas; however, studies have shown that the air Mn levels were not significantly increased after the use of MMT [54]. Therefore, a definitive conclusion on the long-term effects of MMT use on human health has not yet been reached. Neuropsychological tests sensitive to subclinical performance deficits within specific cognitive domains, such as attention, memory, psychomotor functions are used in investigations aimed at determining if exposure to Mn at concentrations far below current exposure limits can leads to more acute or chronic syndromes. Population studies showed subtle behavioral effects in workers chronically exposed to airborne Mn in occupational settings [55].

Manganese at high levels can be toxic to a variety of organs, including the liver, cardiovascular system and, most importantly, the brain. On the other hand, although Mn plays a crucial role in the development of fetal, newborn, and children’s nervous systems, exposure to high concentrations during development has also been associated with signs and symptoms of developmental neurotoxicity [56,57]. Manganese crosses the barriers or takes the path of the olfactory tract to reach the target sites in the brain with the highest level in the putamen, caudate nucleus, and globus pallidus [44,58]. The transport of Mn across the cell membrane is not fully understood. It shares the transport machinery with other metals, such as Zn, Fe, Cu, and Ca [59,60]. Manganese deposits itself into nuclei and mitochondria [61]. Mitochondria is a critical target of Mn-induced neurotoxicity, characterized by mitochondria respiratory dysfunction and elevated levels of super oxidative and nitrogen species [62]. Other mechanisms involved in Mn-induced neurotoxicity include neurotransmission disruption, altered neurotransmitter metabolism, reduced Cu levels in the subventricular zone, proteotoxic stress, and inflammation [63,64].

Although Mn-induced neurotoxicity in occupational settings and its clinical form (manganism) are rare nowadays, the investigation of overexposure to Mn through a variety of other sources (for example, parenteral nutrition, infant formula, and vulnerable populations with high Mn absorption rates secondary to Fe deficiency) and the neurotoxic effects have been an ongoing effort as the range between the toxic threshold and the nutritional level is narrow for this metal [65,66]. Human and experimental studies have shown several mutations in the genes that render individuals susceptible to Mn toxicity, particularly for genes involved in the exportation and transportation of Mn [67,68]. Meanwhile, alternative models, such as *C. elegans*, have been increasingly used to investigate mechanisms and genetic modifiers of Mn-induced toxicity and neurodegeneration [69].

## 3. Mn Exposure and Determination in *C. elegans*

For investigations on the neurotoxic effects of Mn, the nematode can be exposed to Mn at different larval stages (L1–L4) or even as adult worms. However, some factors could reduce the bioavailability of metal and its toxicity, such as the thicker cuticle in adult worms compared to the initial stages [70]. Developmental stage can also influence the endpoint of interest and, therefore, this factor should be considered when designing investigations using this model system. Post-embryonic development is very copious, although it is tightly coordinated in a synchronized population. The nervous system is not fully developed until the end of L2, DNA repair mechanisms are not completely evolved before L4, and the reproductive system is not matured until the end of the fourth larval stage [71,72]. Sensitivity to exogenous agents also varies greatly within the larval stages [73].

For investigations of Mn toxicity on *C. elegans*, both acute and chronic exposure to the trace elements might be of interest. Acute toxicity is most commonly tested in liquid medium with different solvents, such as M9 [74], S medium [75], or 85 mM Sodium chloride (NaCl) + 0.1% Tween [76,77,78]. Attention should be paid to the lack of food, as experiments might not be meaningful in long exposure scenarios due to changed metabolism [79].

Long-term exposure is often conducted on agar plates via the nematode food source, *E. coli*, but can also be conducted in S medium [23]. One necessary attention should be concerning *E. coli* as the alive bacteria might be able to metabolize the compounds of interest, and its inactivation before exposure should be considered. Manganese can be added to the bacteria LB (Luria broth) medium before or after the inactivation of the *E. coli* (for example, via heat) [39,80]. Even when working with inactivated *E. coli*, one should investigate if no changes in Mn-species occur due to the experimental settings, especially when focusing on species-specific effects. The exposure time should be taken to determine how the experiment should be set out depending on the specific research interest. The applied Mn species is one parameter to be considered next to the method of exposure, incubation time, and larval stage. Commonly, Manganese chloride (MnCl_2_), as described in Avila et al., or manganese sulfate (MnSO_4_) are incubated [81,82]. Both of these species are Mn (II) compounds, which is Mn’s most stable oxidation state.

The different exposure methods can lead to great variances In absorption efficacy and bioavailability. The effective interpretation of novel results and comparisons of these with results from other studies requires that attention be paid to bioavailability and concentration–response or concentration–effect curves of Mn which need to be tested for each application method and developmental stage. Manganese toxicity varies strongly between metal species, administration form, and exposure time [83,84]. For the determination of concentration–effect, survival and lifespan assays are standard procedures in *C. elegans* laboratories. Verifying the lethal dose at which 50% of the animals die (LD50) for a specific Mn-containing compound is a crucial first step in the processes of determining a suitable exposure concentration for subsequent investigations of a desired novel endpoint [85]. Mutagenicity and genotoxicity are usually examined after sub-toxic Mn exposure, and lifespan assays are often conducted to investigate changes after slight chronic overexposure or depletion of the trace element [83].

As some substances exhibit poor absorption efficiency, the applied dose does not necessarily reflect the effective internal dose within the worms. To understand the actual toxicity, it is, therefore, necessary to measure the Mn content in *C. elegans*. The absorbed dose can be measured in a pool of worms, followed by normalization to either worm number or protein content. The latter might be more reliable, as inaccuracies in worm number might easily appear, but the protein content is precisely determined in each worm. The most common method for trace element measurement is the analytical approach via inductively coupled plasma–mass spectrometry (ICP–MS) [39,81]. Even though this method is very sensitive, it can only be used for total elemental content analysis, as samples have to be destroyed. Two other commonly used analytical approaches for Mn quantification are atomic absorption spectroscopy (AAS) and inductively coupled plasma–optical emission spectrometry (ICP–OES), which can also be applied to *C. elegans* samples [86,87]. The group of Michalke has extensively studied the Mn species within the brain of rats and human serum/cerebrospinal fluid [88,89], but no such studies have been conducted on the nematode as of yet. Such investigations have been carried out in *C. elegans* for other trace elements, such as selenium [90], but Mn species in the worm still need to be understood further. Therefore, we cannot make any statements regarding different species-specific toxicities, transversions, and metabolic routes.

For localization of metal abundance, histochemical staining is utilized for *C. elegans* but lacks sensitivity (there are no specific stains for Mn) and quantitative data. Using event-mode X-ray fluorescence microscopy, McColl et al. were able to successfully visualize Mn on a subcellular level [91]. This scanning method, which rapidly accelerates acquisition rate and decreases radiation damage compared to traditional X-ray fluorescence microscopy, made visualization of elemental compartments and quantitation of Mn in nematodes possible [92,93,94,95]. Another method for intra-body localization of Mn is the visual and qualitative laser ablation inductively coupled plasma–mass spectrometry (LA-ICP-MS) [96,97].

## 4. Exploring Manganese Toxicity in *C. elegans*

### 4.1. Neurotoxicity Assessments

Manganese toxicity has been plentifully investigated in *C. elegans*. This is mainly due to the conservancy of target genes to human diseases and the ease of genetic manipulation that allows for mechanism-based studies. Environmental exposure to Mn is linked to a neurodegenerative condition named manganism, whose pathophysiology resembles Parkinson’s disease. The main target of Mn is the dopaminergic (dAergic) system which can be accessed both in vivo and in vitro in *C. elegans* through observing neuron morphology, behavioral changes, neurotransmitter determination, and related-genes expression. The dAergic neurons represent 40,000–45,000 of the total neurons in rats, 160,000–320,000 in monkeys, and 400,000–600,000 in humans [98]. Alternatively, *C. elegans* offers a simple and easy evaluation by fluorescent transgenics, which provides evidence that there are only eight dAergic neurons in the hermaphrodites, which are divided into three classes based on their morphology, as follows: four CEP neurons innervate the tip of the nose (two ventral, the CEPV, and two dorsal, the CEPD), two ADE neurons innervate the head cuticle, and one pair of posterior PDE neurons are located in the posterior cuticle. Since males play the coupling function, six additional dAergic neurons are located in the tail [99]. Constructions combining the fluorescent label green fluorescent protein (GFP) and the sodium-dependent DA transporter DAT-1 expressed in dAergic neurons in *C*. *elegans* represent the first choice [33,100]. The main strains include BZ555 (egIs1 [dat-1p::GFP]), BY200 (*vtIs1* (P*dat-1*::GFP, pRF4(*rol-6*(su1006)), and BY250 [*vtIs7*; P*dat-1*::GFP]. Another option is the strain OH7547 (otIs199 [cat-2::GFP + rgef-1(F25B3.3)::DsRed + *rol-6*(su1006)].), that expresses GFP in dAergic neurons and dsRed pan-neuronally [101]. However, it should be considered that there are intrinsic differences in each strain in response to exposure to the compounds [102].

Endpoint characterizations of Mn-induced neurotoxicity include neuronal development, puncta, neuronal absence or shrinkage, neuronal gaps, absence of cell bodies, and reduction in fluorescence intensity [18,102,103]. Such an analysis is feasible by using images acquired with confocal or fluorescence microscopy. As the neurons are distributed in *C. elegans* cylindrical body, Z-stack should be considered for representative images and, for that, worms can be anesthetized with 0.2% tricaine/0.02% tetramisole, 1 mM levamisole, or 10–25 mM *sodium Azide* (NaN3). Although it is not used as frequently as fluorescence, electron microscopy can also be used. The sum of slices and optical densitometry can be analyzed using the ImageJ software (NIH) (Figure 1).

All the main neuronal systems have been considered under Mn exposure (GABA, Ach, 5-HT, Glu), although Mn selectively affected *C. elegans* dAergic neurons [18]. The crucial mechanisms implicated in Mn neurotoxicity are discussed below.

### 4.2. The Benefits of C. elegans Genetic Conservation on Mn Toxicity

Investigations looking at the adverse effects of exposure to Mn are generally conducted in wild-type animals stratified by age and other phenotypic characteristics. Accordingly, Mn toxicity was associated with the reduction in *C. elegans* survival and deficits in lifespan that depend on the concentration of Mn and larval stage [8,104,105]. In addition, aged animals exposed to Mn are more susceptible to Mn toxicity when compared to young animals, which is mainly related to protein homeostasis impairments [106].

The molecular basis of Mn toxicity is readily assessed by looking at proteins responsible for DA biosynthesis, packaging, and reuptake which, are all present in *C. elegans* [99] (Figure 2). The DA biosynthesis requires tyrosine, which is firstly hydroxylated by tyrosine hydroxylase (TH, *cat-2*) to produce L-Dopa. Dopa decarboxylase (*bas-1*) converts L-DOPA into DA [107]. Thus, DA interacts with different receptors encoded by the four genes *dop-1*, *dop-2*, *dop-3*, and *dop-4* in *C. elegans*. Furthermore, *dop-1* and *dop-4* are analogous to mammals’ D1-like receptor genes and *dop-2* and *dop-3* to D2-like receptor genes [108]. Additionally, *dop-5* and *dop-6*, have been proposed to be DA receptors based on sequence similarity to *dop-3*. After release, the DA efflux from the synaptic cleft moves through the conserved DA transporter (*dat-1*), followed by DA packing dependent on the vesicular monoamine transporter (*cat-1*) [108]. In *C. elegans*, Mn selectively impairs the dAergic neurons rather than other neurons, which is implicated in *dat-1* since its mutation (knock-down) results in hypersensitivity to Mn [18].

The toxicity of Mn has also been related to an increase in reactive oxygen species (ROS), a reduction in glutathione (GSH) levels, and mitochondrial changes [109,110,111]. The DA extracellular levels are also crucial for Mn-induced oxidative stress in *C. elegans*. An NADPH dual-oxidase, BLI-3, potentiates the synthesis of ROS from DA-derived toxic species, which can be transported via uptake through the DA transporter DAT-1. On the other hand, the *bli-3* loss caused hyper-resistance to Mn toxicity [18]. Moreover, the Mn-induced oxidative stress can be protected by pre-treatment with anti-oxidants, such as ebselen [112].

Within cells, Mn accumulates predominantly in mitochondria, especially as Mn (II). It implies changes in oxidative respiration, resulting in high ROS production and culminating in mitochondrial dysfunction [113,114]. In *C. elegans*, Mn exposure significantly alters Fe and Ca homeostasis, combined with mitochondria and endoplasmic reticulum dysfunction and impairments in protein homeostasis, which may further confer the compound toxicity [82,106,110].

The genetic machinery implicated in metal homeostasis and transport is preserved in *C. elegans*. Three homologs for the divalent metal transporter (DMT1) have been characterized in *C. elegans*, namely *smf-1*, *smf-2*, and *smf-3*. Here, SMF-3 is the primary Mn uptake transporter in the worm, whereas SMF-1 has a minor role in this process, and SMF-2 regulates metal content (it is also responsible for Fe uptake) [115]. In mammals, DMT1 expression mostly occurs in the proximal duodenum, kidney, and brain tissues, where it plays the non-selective metal transport role [59]. In *C. elegans*, SMF-1 is mainly expressed in the intestine and associated gland cells, SMF-3 is expressed along the intestine and weakly in epidermal and neural cells, and SMF-2 is restricted to the epithelial cells of the pharynx and the pharyngeal-intestinal valve cells, and gonad sheath cells at the adult stage [115]. Mutations in *smf-1* and *smf-3* have been shown to confer increased Mn tolerance, while mutation in *smf-2* led to increased Mn sensitivity. In addition, the inflammatory response to pathogens is also affected by the mutation in *smf* transporters which has been associated with the Mn role in the innate immune system [116]. The P-type Ca^2+^/Mn^2+^ ATPase (PMR1) is also implicated in the transport and homeostasis maintenance of Mn in the worm. Here, *pmr-1* reduced expression by mutation, and RNAi rendered worms resistant to oxidative and pathogenic stress, respectively [116,117].

In *C. elegans*, the loss of heat shock proteins *hsp-70* intensifies the protein oxidation and the dAergic neuronal degeneration through inhibition of the transcriptional upregulation of *PINK1*, a gene that has, in fact, been linked to Parkinson’s disease [82]. In addition to *pink1*, most related Parkinson’s disease genes, including *park2/parkin, dj-1*, and *lrrk2*, have at least one *C. elegans* homolog [118,119,120,121]. The role of *dj-1* in oxidative stress regulation is an intersection point associated with both PD and Mn toxicity [122]. The influence of *dj1* mutation over Mn uptake has been investigated in two different homologs present in *C. elegans*, namely *djr1.1* and *djr1.2*, in the nuclear location and cytosol of head neurons, respectively. Manganese uptake was increased in the *djr1.1* deletion mutant, while in double mutant *djr1.1*; *djr1.2*, Mn was not dispersed inside the worm [109].

The effects of *pdr1* and *djr1.1* on modulating Mn transport were evaluated in the transgenic expressing human α-synuclein protein (α-Syn). Here, α-Syn is a small protein whose accumulation of its misfolded form is also implicated in Parkinson’s disease neurodegeneration [123]. *Caenorhabditis elegans* transgenic expressing α-Syn and exposed to Mn presented a reduction in both Mn accumulation and Mn-induced oxidative stress when combined to a background of *pdr1* and *djr1.1* loss [109].

In addition, djr-1.2 has been reported for mitigating Mn-dependent lifespan reduction and DA signaling alterations conditioned on the DAF-2/DAF-16 signaling pathway [124]. Moreover, the absence of *pdr-1* (*parkin* homolog) enhanced Mn accumulation and dAergic neurodegeneration in the worm, which was related to defects in exporting Mn through the putative Mn exporter ferroportin (*fpn-1.1*) [109,125].

The conservation of the transcription factor homolog of mammalian Nrf2 (SKN-1) in the worm confers protection on Mn-induced neurotoxicity when overexpressed or activated in the nucleus [18]. In *C. elegans*, SKN-1 is upstream regulated by DAF-2 (a receptor similar to the mammalian insulin-like growth factor 1, or IGF-1, receptor) pathway. It is well recognized in the worm that DAF-2 activates AGE-1 (a homolog of phosphatidyl inositol-3 kinase, PI3K), which converts phosphatidyl inositol-2 phosphate (PIP2) to PIP3, which is responsible for the phosphatidyl inositol dependent kinase (PDK-1) activation. For instance, PDK-1 activates both AKT-1/2 and SGK-1 which retains SKN-1 and DAF-16 transcription factors in the cytoplasm (inactivated). Under Mn exposure, the AKT-1 expression is increased, and both SKN-1 and DAF-16 remain cytoplasmatic [105,126].

### 4.3. Mn Impairments on C. elegans Behavior and the Relationship to Its Toxic Mechanism

In mammals, neurotransmitters and hormones are the chemical messengers whose unbalance directly impact behavioral responses. One of the factors which motivated Sydney Brenner, who proposed *C. elegans* as an experimental, was that the animal had fewer neurons than a fruit fly. Later, Dr. Brenner was able to verify that gene mutations and neuron changes could be reliably measured through *C. elegans* behavioral observation. Since DA is the main target of Mn, we made some comments on its influences on *C. elegans* behavior.

Dopamine signaling modulates a range of behaviors in the worms, such as locomotion [127] and learning [128,129]. In addition, it allows animals to efficiently search for new sources of bacteria and stay at the food site. Additionally, DA-mediated slowing of locomotion is assessed by a Basal slowing response test, in which locomotion rates (on bacteria) are compared in well-fed worms transferred to bacteria-seeded plates vs. those placed on plates without bacteria [127]. Deficits in the dAergic system are made manifest by a reduced basal slowing response (which do not slow when finding the food) in worms exposed to Mn [105].

*Caenorhabditis elegans* can respond to environmental stimuli through sensory systems and modify its behaviors, making it a suitable model for studying behavioral plasticity. Ethanol preference is predictive of functional DA and 5-HT development, whereas worms with losses in DA and 5-HT biosynthesis (*cat-2* and *tph-1* mutants) do not show it [130]. The overload of Mn results in death in dAergic neurons due to modifications in locomotor activity in *C. elegans*, as observed in humans affected by manganism [127,131]. Tracker systems connected to a stereoscope can precisely indicate the distance traveled, speed, acceleration, and minor changes in locomotion [132,133]. However, body bends can be counted on plates without the bacterial lawn. Manganese reduces the body bends, mainly when the exposure occurs in the L1 larval stage [134].

The feeding behavior is examined in toxicological approaches and can be evidenced by the counting of pharynx rhythmic contractions in *C. elegans*. Pharyngeal pumping is sustained under the integrated output of the nervous system, including the glutamatergic, GABAergic, serotoninergic, and cholinergic’ systems [135,136]. In this regard, 5-HT is released in the presence of food and gives rise to glutamatergic activation, which contributes to high pharyngeal pumping rates [137]. Additionally, DA impacts pharyngeal pumping by acting on the dAergic receptor DOP-5 inducing 5-HT release [138]. This fact can justify the decreasing pharyngeal pumping rate under manganese exposure, as well as in *cat-2* mutant worms with reduced DA production [8,18,139].

The reproductive profile can be assessed in *C. elegans* by counting the eggs inside the uterus (egg production), the eggs laid on the plate during a time period (egg laying), the hatching rate (brood size), and male mating behavior. Manganese delayed the reproduction in the worm by transient reducing the egg production and egg laying [7].

Thus, worm behaviors are strongly influenced by changing the genome expression and neurotransmitters’ homeostasis. In addition, behavioral observations add important information, mainly by pointing out the principal genetic targets, thus, simplifying the investigation of Mn toxic mechanisms. Figure 3 shows the effects and some signaling pathways that Mn may affect.

## 5. *C. elegans* Model to Study Neurodegenerative Disease

A multitude of neurodegenerative diseases, such as Parkinson’s disease (PD), Alzheimer’s disease (AD), Huntington’s disease (HD), and amyotrophic lateral sclerosis (ALS), are characterized by disruption in metal homeostasis and subsequent accumulation of disease-related protein aggregations [140,141,142]. In exploring Mn toxicity in *C. elegans*, we see a relationship between Mn and many neurodegenerative diseases.

Parkinson’s disease is pathologically characterized by Lewy bodies, composed mainly of α-synuclein. Anatomically, we also see degeneration of the dopaminergic neurons projecting from the substantia nigra resulting in the three cardinal clinical features of PD, which are rigidity, resting tremors, and bradykinesia [51,143]. Manganese-induced Parkinsonism presents similarly to idiopathic Parkinson’s disease with motor deficits, such as bradykinesia, tremors, difficulty walking, and facial vacancy. Similarly, both conditions are characterized by degeneration of dopaminergic (DAergic) neurons as well [50,51]. However, despite these similarities, there are distinct differences in the pathological presentations of each condition.

In Mn-induced Parkinsonism, we note degeneration of dopaminergic (DAergic) neurons mainly in the substantia nigra pars, but in idiopathic Parkinson’s we see the same degeneration in more of the basal ganglia, including the substantia nigra, globus pallidus, subthalamic nucleus, and striatum [51,140]. Furthermore, we also see the lack of Lewy bodies in Mn-induced Parkinsonism. However, Mn still induces aggregation and misfolding of alpha-synuclein and there exists emerging evidence to suggest that these accumulations are responsible for the neurotoxicity in Parkinson’s disease [140]. While the worm has no α-synuclein orthologue, *C. elegans* models for PD are based on transgenic worms overexpressing wild-type or mutant forms of the human α-synuclein [144]. Further studies aiming to better understand the pathophysiology of these similar, yet different, diseases may be helpful in developing therapeutic strategies for both disorders [51]. Recently, it was reported that metal dyshomeostasis is further induced in a parkin-deficient nematodes (PD model), where metallothionein 1 acts as a potential protective modulator in regulating homeostasis [84].

On the other hand, AD is characterized by several key pathologies, with one of them being AB peptides. It is demonstrated by Tong et al. (2014) that increased exposure to Mn is correlated with increased plasma amyloid-β (Aβ) peptides, mini-mental state examination score, and clinical dementia rating scale score. The study suggests that Mn induces Aβ peptide-related cognitive impairment by disrupting Aβ peptide degradation mechanisms [145]. However, we also note a meta-analysis of 17 studies that suggests Mn deficiency may be contributory to AD development [146]. While the relationship between Mn and AD is still not clear, there is evidence to suggest that disruption in Mn homeostasis may play a pathogenic role in the development of AD.

There are several reasons as to why *C. elegans* is an exceptional model for neurodegenerative diseases. This model has allowed for extensive studies into PD, including links between toxic compounds exposure, such as Mn, and increased risk of PD [21,144]. While in AD we do have an amyloid precursor protein (APP) orthologue, it lacks Aβ homology to humans. Thus, to measure Aβ toxicity in AD, transgenic expression of recombinant Aβ is used to circumvent *C. elegans* APP mechanisms [147]. These Aβ peptides can induce behavioral changes in *C. elegans*; for example, the strain CL2006 induces Aβ deposits in body wall muscles, resulting in rigidity and paralysis. Thus, paralysis in this scenario can be used as a quantifiable behavioral output of Aβ toxicity. Further evaluation of Aβ-mediated neurodegeneration can also be quantified through visualization of glutaminergic neurons in the tail of the worm [147]. *Caenorhabditis elegans* can be used as a model for a number of other pathologies, such as tauopathy in AD [147] or as a PDR-1 model in PD [144].

Neurodegenerative diseases, such as AD and PD, are multifactorial diseases with numerous complex pathologies. While there are multiple studies on the effects of Mn in AD and PD, how Mn contributes to subsequent protein aggregation and neurotoxicity has yet to be well understood [148]. The simplicity and homology of *C. elegans* to human have proven to be useful in the understanding of these neurodegenerative diseases, and further studies using *C. elegans* may be beneficial in clarifying this relationship. Translating some of the insights made in *C. elegans* maybe helpful in developing therapeutic approaches to AD, PD, and other neurodegenerative diseases in the future.

## 6. Concluding Remarks

Given the chronic environmental exposure and irreversible neurotoxicity of Mn, future studies should utilize contemporary methodologies to better characterize its mechanisms of neurotoxicity. Moreover, combined exposure to Mn and other divalent metals (and/or xenobiotics) can be addressed in future studies, since it is plausible that divalent metals can influence each other.

Finally, the corroboration of the mechanisms of Mn neurotoxicity between the nematode and mammals, the ease of manipulation, genetic conservancy, and the short-time of observation positively implicate *C. elegans* as a complementary and alternative biological model to address the complex signaling pathways associated with both the essentiality and neurotoxicity of this metal. Findings linking Mn toxicity with molecular targets provide critical information which will likely identify putative target sites for pharmacological interventions, not only for Mn intoxication but also in Parkinson’s disease treatment by taking into consideration the pathophysiological similarities between the two conditions.

## Figures and Tables

**Figure 1 biomolecules-12-01396-f001:**
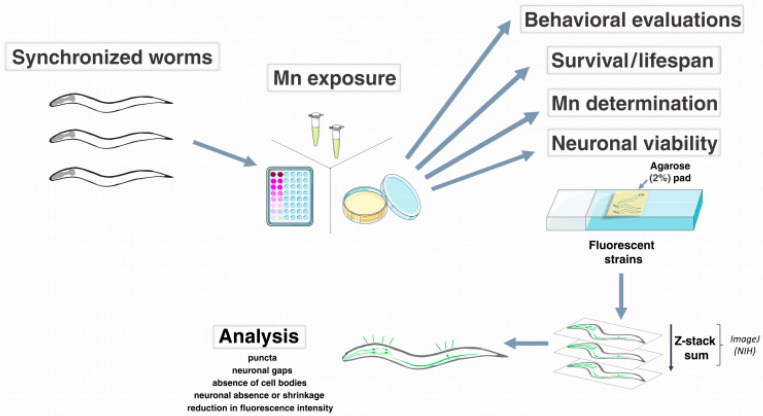
Worms at the same larval stage (L1 to L4, or even adult worms) can be exposed for minutes or hours in liquid medium (96-wells or microtubes) or NGM dishes. Afterwards, parameters, such as behavioral evaluations, survival/lifespan, Mn determination, and neuronal viability, can be evaluated [33]. The Mn neurotoxicity is observed in fluorescent strains (BZ555 (egIs1 [dat-1p::GFP]), BY200 (vtIs1 (Pdat-1::GFP, pRF4(rol-6(su1006)), and BY250 [vtIs7; Pdat-1::GFP] [102]). Images are acquired in Z-stack, summed, and analyzed in ImageJ (NIH). The presence of puncta, neuronal absence or shrinkage, neuronal gaps, absence of cell bodies, and reduction in the fluorescence intensity are evidence of neurotoxicity [18,103].

**Figure 2 biomolecules-12-01396-f002:**
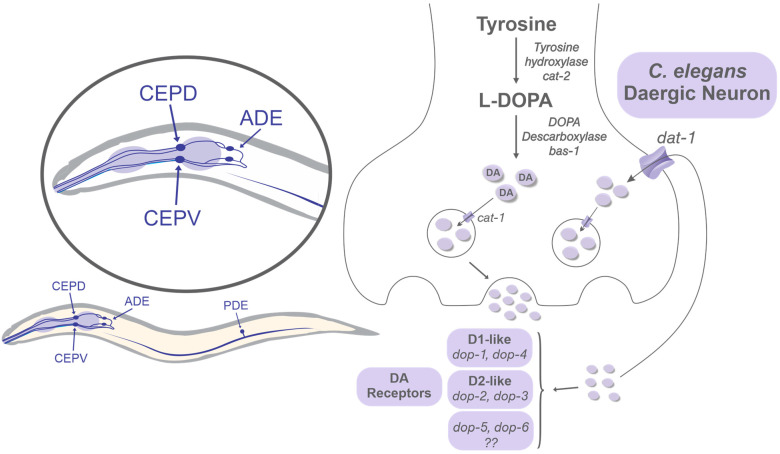
dAergic conservancy in *C. elegans*.

**Figure 3 biomolecules-12-01396-f003:**
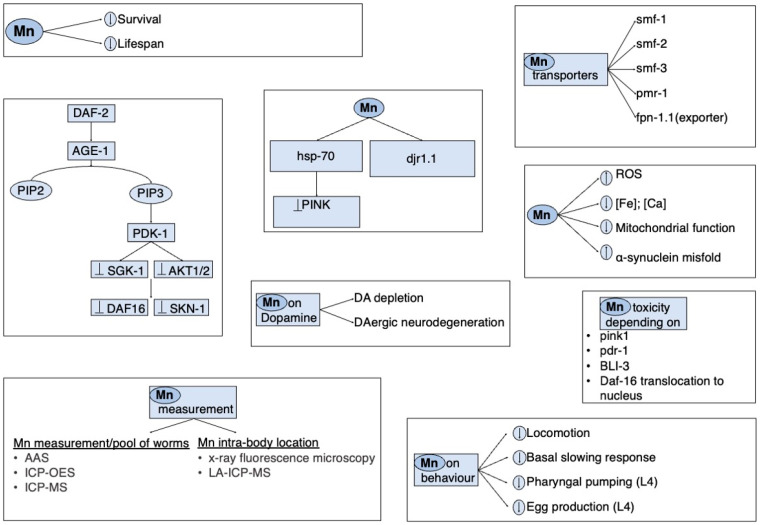
Key targets of Mn on its neurotoxic effects in *C. elegans*.

## Data Availability

Not applicable.

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
