# Peer review of "Caenorhabditis elegans as a Model to Study Manganese-Induced Neurotoxicity"

_biomolecules, 2022, doi:10.3390/biom12101396_

Round 1

Reviewer 1 Report

Overall interesting manuscript with nice figures.  This reviewer recommends that the entire manuscript be carefully proof read and educate for grammar and clarity.  I have attached a document with recommended changes but this is not exhaustive list. 

Author Response

Thank you very much for your comments. We have added all your suggestions. They are highlighted in yellow in the manuscript. 

Reviewer 2 Report

This is an interesting review of the use of C. elegans as an alternative model for neurotoxicity studies. The review focusses on the neurotoxicity of manganese, and summarizes studies done in the worm with this metal. The authors are leaders in this field and provide an extensive discussion of the issues and an exhaustive bibliography. Of note is that in addition to "traditional" information on manganese neurotoxicity, novel state-of-the-art insights and approaches are presented.

Author Response

Thank you very much for your comment. 

Reviewer 3 Report

Is a very good and interesting review. The article is clear and the authors review and highlight the interest of using C. elegans as a model to study Mn neurotoxicity. As Mn seems to be involved in several degenerative diseases like PD or AD; the authors propose C. elegans as a good model to study these human diseases. They expose different methods to measure Mn in C. elegans and some important points to consider when using C. elegans (genetic, developmental stage, feeding conditions…)

Only, I noted some minor remarks

Comments:

Line5:    the last name is not well written.

line67: it would be helpful to give a reference on who stablished the homology between the genomes.

line69: is the reference 19 the correct one?

line71: Can the authors cite the original article here? The one where the absence of certain organs found in mammals is described?  I am not sure that reference 20 is the most appropriate.

Line83: the reference 26 is about mosses, not C. elegans.

Line90: the reference 33 is not related with C. elegans.

Line93: the authors forgot to add references for zinc and mercury

Line139; the reference 54 is not related with to Mn accumulation in fetal, newborn, or children’s nervous system, but related with Mn and golgi apparatus in cells. Can the authors change the reference?

Line194: the reference 82 is not related to Mn but to the maintenance of C. elegans.

Line200: write the reference before the point.

Lines220-228: the references 95-97 are not related to C. elegans imaging. In the bibliography, there are a few publications were x-ray fluorescence is used to image metals in C. elegans, either using tomography or not. The authors should cite here articles were x-ray fluorescence have been used to image metals C. elegans; not general articles on the use of x-ray fluorescence in biology.

Line245-252: Can the authors revise this paragraph? and include a link for each strain, as they did for BZ555 ?

I could not find BY200 or BY250 at https://cgc.umn.edu/

A number was not given to reference Nass et al., 2002

I think that, in line 249, authors forgot the strain name for the genotype otls199.

Line 292 is in bold, is it normal?

Line 333: I found that the reference 125 is a general one, about genes database, and not directly related to djr1.1, djr1.2 ? Can the authors cite a specific one?

Lines349-355: a sentence is in bold but no reference is given on this statement.

I saw that reference 129 refers to AKT1,2, DAF-16 and cytosol accumulation, but I could not see SKN-1 in, do the authors have another reference to include here?

Line 375: the reference 134 is related with Mn and Fe in rats. I do not think it should be cited in this article. The review should continue to be focus on Mn and C. elegans.

Line386: are the underline word, link and reference 141 correclty placed?

Figure3. Some missing references in ‘Mn on dopamine’ and ‘Mn toxicity depending on’.

The reference for ‘Mn measurement, intra-body location, x-ray fluorescence microscopy’ is not correct. Authors should cite a reference here where C. elegans have been analyzed by x-ray fluorescence.

Why are some references in blue and others in green?

Should the references be listed using the author and year or using numbers?

There are two legends in figure 3, before and after the figure.

Line404. The reference 143 is related to PD and AD. Do the authors have other references for HD and ALS?

Lines414-416. The authors inversed the terms Mn-induced Parkinsonism-like and idiopathic Parkinson’s.  Do the authors have another reference instead of 143 that fits better indicating the differences between PD and parkinsonism? The reference 143 is related to AD and PD.

Line434. Here authors could cite the reference of the original article (Du K et al. 2007), rather than the reference of a review article.

Line443. For clarity, can the authors write what is CL2006, a strain?

Author Response

Thank you very much for your comments. We  have attached our replay. 
